# Is Prostate Urethral Lift Effective in Patients with Multiple Comorbidities?

**DOI:** 10.3390/jcm11071928

**Published:** 2022-03-30

**Authors:** Sun-Tae Ahn, Dong-Hyun Lee, Sun-Bum Cho, Hyun-Soo Lee, Da-Eun Han, Tae-Yong Park, Du-Geon Moon

**Affiliations:** 1Department of Urology, Korea University Guro Hospital, No. 148, Gurodong-ro, Guro-gu, Seoul 08308, Korea; asturology@gmail.com (S.-T.A.); donghyunlee13@gmail.com (D.-H.L.); rhfughfkddl@naver.com (S.-B.C.); nny9405@gmail.com (H.-S.L.); yahanda1026@gmail.com (D.-E.H.); 2Department of Urology, Uijeongbu Eulji Medical Center, Eulji University, Dongil-ro 712, Uijeongbu-si 11759, Korea; ptyurology@gmail.com

**Keywords:** prostatic urethral lift, LUTS, comorbidity

## Abstract

A prostatic urethral lift (PUL) can be performed under local anesthesia in patients normally at high risk for general anesthesia due to multiple comorbidities. However, the clinical efficacy of PULs in patients with multiple comorbidities remains unknown. Therefore, in this this study, we aimed to evaluate the clinical efficacy of the PUL in patients with a high number of comorbidities by comparing its clinical efficacy in these patients with that in healthy individuals. We performed a retrospective observational cohort study, in which patients who underwent a PUL between December 2016 and January 2019 at a single tertiary care center were categorized into two groups: healthy individuals who wanted to preserve sexual function (Group 1) and patients with a high number of comorbidities who were at high risk for general anesthesia, based on an American Society of Anesthesiologists (ASA) score of ≥3 (Group 2). The International Prostate Symptom Score (IPSS), maximum urinary flow rate (Qmax), and post-void residual urine (PVR) were obtained preoperatively and compared throughout the 2-year follow-up. A total of 66 patients were enrolled, of whom 36 patients were included in Group 1 and 30 in Group 2. In Group 1, IPSS, IPSS quality of life (QoL), and Qmax significantly improved and were then maintained during follow-up, whereas, in Group 2, improvements in these parameters were not maintained during follow-up, except for IPSS QoL. Eleven patients (36%) in Group 2 required additional treatment for the recurrence of lower urinary tract symptoms. In conclusion, patients with a high number of comorbidities had a low therapeutic effect after PUL, suggesting a high rate of treatment failure. Therefore, comorbidity status should be considered when evaluating the potential benefits of the PUL procedure during preoperative counseling.

## 1. Introduction

South Korea has been categorized as an “aged society” since 2000 [1]. The prevalence of many health problems, including issues with the urinary system such as lower urinary tract symptoms (LUTSs), is associated with increased age. According to a population-based survey, the prevalence of LUTSs increased from 78% among men in their 40s to over 90% among men aged >60 years [2]. LUTSs in older men are usually associated with urinary outlet obstruction due to progressive enlargement of the prostate gland, leading to benign prostatic hyperplasia (BPH). Pharmacotherapy using alpha-blockers or 5-alpha reductase inhibitors (5-ARI) is effective and commonplace in managing moderate to severe LUTSs, and anticholinergic agents or beta-3-agonist may be offered to patients with moderate to severe predominant storage LUTS [3,4]. However, a large proportion of patients are non-compliant due to insufficient relief or adverse side effects caused by these drugs [5,6,7]. Transurethral resection of the prostate (TURP) is the gold standard surgical approach for BPH treatment. It is well-established that TURP is effective in improving LUTSs and maintains its efficacy in the long-term [8]. Other surgical alternatives to TURP have been introduced, including holmium enucleation of the prostate (HoLEP), photoselective vaporization of the prostate (PVP), and bipolar transurethral resection in saline [9,10,11]. However, most conventional transurethral procedures have several disadvantages. TURP especially is associated with significant peri-operative morbidity and long-term complications, including bleeding requiring transfusions (2.9%), transurethral resection syndrome (1.4%), urethral stricture (7%), and urinary incontinence (3%) [3,4,12]. Furthermore, these transurethral procedures are frequently associated with an increased risk of sexual dysfunction, particularly ejaculatory dysfunction [12,13,14]. In addition, there is a risk associated with anesthesia in patients with multiple and/or severe comorbidities. Therefore, minimally invasive treatments that are feasible via local anesthesia, such as prostatic urethral lift (PUL) and prostatic embolism, have been introduced [15].

The PUL has attracted considerable attention as a minimally invasive alternative since 2011, when the first study was conducted on 19 men in Australia [16]. In South Korea, the Korean National Evidence-based Healthcare Collaborating Agency (NECA) approved the use of UroLift as a novel medical technology in 2015. Several studies have provided promising results, including a 2020 study on the Korean population [17]. However, the clinical efficacy of the PUL in patients with a high number of comorbidities remains unclear. Moreover, mid-term outcomes at more than a year’s time have not been assessed in Korean men. Therefore, we aimed to evaluate the clinical efficacy of the PUL in patients with a high number of comorbidities by comparing it to that in healthy individuals, while reporting mid-term outcomes.

## 2. Materials and Methods

### 2.1. Ethical Statement

This study was approved by the Institutional Review Board of Korea University Guro Hospital and was conducted based on the ethical standards of the 1964 Declaration of Helsinki and its later amendments. Preoperatively, all patients were informed of the surgical treatment options, and written informed consent was obtained from all study participants.

### 2.2. Study Design and Patients

This retrospective observational cohort study was conducted at a single tertiary care center in South Korea. A PUL was primarily offered to patients who were motivated to preserve ejaculatory function. As other minimally invasive treatments such as Aquablation and Rezum have not yet been approved in Korea, PUL was the only procedure that could be offered to patients who wanted to preserve ejaculatory function. The participants were those who wished to receive treatment for refractory LUTSs and who were either motivated to preserve ejaculatory function or were classified as high-risk for general anesthesia. We divided the patients into two groups: healthy individuals who wanted to preserve ejaculatory function and patients at high risk for general anesthesia due to a high number of comorbidities. The high comorbidity group was defined as patients at high risk for general anesthesia based on an American Society of Anesthesiologists (ASA) score of ≥3. Other inclusion criteria were as follows: age ≥ 50 years, International Prostatic Symptom Score (IPSS) ≥ 9, prostate volume between 15 and 100 mL, and prostate specific antigen < 10 ng/mL. Patients with active urinary tract infections were excluded. Before each prostatic urethral lift procedure, all patients underwent a transrectal ultrasound and cystoscopy to assess prostatic and urethral structure for eligibility for surgery. Patients taking alpha-blockers were asked to suspend these drugs 1 week before pretreatment evaluation.

### 2.3. Surgical Procedure, Follow-Up, and Outcomes

The Urolift prostatic urethral lift system (Urolift^®®^; Neotract Inc., Pleasanton, CA, USA) allowed for the transurethral insertion, using cystoscopy, of small permanent implants to compress the prostatic tissue, thereby treating the urethral obstruction. The number of implants varied depending on adenoma size and the nature of the obstruction. All procedures were performed as previously described [16], by a single surgeon (DGM), with the patient under either intravenous sedation using remifentanil and propofol, or local anesthesia using a 50 mL intravesical lidocaine injection and 2% lidocaine gel on the urethra. A urethral catheter was inserted and later removed on postoperative day 1 before discharge.

Patients followed up with visits at 1 week, 1 month, 3 months, 6 months, 1 year, and 2 years post-procedure. All patients underwent a preoperative assessment, including the International Prostate Symptom Score (IPSS), IPSS Quality of Life (QoL), a male sexual health questionnaire to assess ejaculatory dysfunction, short-form version (MSHQ-EjD), and a uroflowmetry test (maximum urinary flow rate: Qmax, post-void residual urine [PVR]). These parameters were assessed during follow-up visits. Changes in the measures of these parameters from the baseline were compared across the treatment groups at each follow-up visit. If the patient underwent additional surgery or treatment during follow-ups, the results before the new treatment were used for the analysis. All complications were assessed during follow-up.

### 2.4. Statistical Analysis

The baseline characteristics of the patients are descriptively analyzed. Categorical variables are reported as numbers and percentages, while continuous variables are reported as mean ± standard deviation (range). The changes in baseline characteristics and outcomes at multiple time points were analyzed using the paired Student’s *t*-test or Wilcoxon signed-rank test, based on the distribution of the paired data. Thereafter, we used a generalized linear mixed model to examine whether there was a trend difference in symptom improvement, including IPSS, IPSS QoL, Qmax, and PVR, based on time after PUL in both groups, adjusting for age, prostate volume, and number of implants. The chi-square test or Fisher’s exact test was used to test for significant differences in adverse events. SPSS version 26.0 (IBM Corp. Released 2011, IBM SPSS Statistics for Windows, Version 20.0, Armonk, NY, USA) was used for statistical analyses. Statistical significance was set at *p* < 0.05.

## 3. Results

Sixty-six patients were included in this study, of whom thirty-six were classified as healthy individuals who underwent a PUL in order to preserve ejaculatory function (Group 1) and thirty as patients with a high number of comorbidities, who underwent a PUL under local anesthesia (Group 2). The baseline characteristics of the two groups are summarized in Table 1. There are significant differences in patient age and comorbidities based on group classification characteristics; however, there are no differences between the groups regarding prostate volume, symptom severity, and deployed implants (Table 1).

Significant improvements in total IPSS and IPSS QoL were observed in both groups throughout the 2-year follow-up period (Table 2). In Group 1, the Qmax maintained this improvement after 2 years of follow-up, whereas in Group 2, improvement was not maintained after 6 months. Additionally, improvements in total IPSS, IPSS QoL, and Qmax in Group 1 were superior to those in Group 2 at all time-points, except for Qmax at 1 month (Table 2). In both groups, there was no significant decline in ejaculatory and sexual parameters (assessed by IIEF and MSHQ scores) over the course of the 2-year and 1-year follow-up, respectively (Table 2). Over the 2-year follow-up, the level of improvement in terms of IPSS (*p* < 0.001), IPSS QoL (*p* < 0.001), and PVR (*p* = 0.008), was better in Group 1 than in Group 2 (Figure 1). However, these trends were not observed for Qmax (*p* = 0.209) (Figure 1).

Over the course of the 2-year follow-up, three (8.3%) patients in Group 1 and eleven (36%) in Group 2 required additional treatment for a recurrence of LUTSs (Table 3). Additional treatments included surgical reintervention (TURP or PUL), the use of an indwelling catheter, and the administration of medication (alpha-blockers or 5 alpha reductase inhibitors). The treatment failure rate was significantly higher in Group 2 than in Group 1 (*p* = 0.007). Other frequently observed adverse events in both groups were dysuria and hematuria, and no significant differences were identified between the groups (Table 3). There were no reported cases of retrograde ejaculation after PUL in either arm. One patient presented with prostatic abscess and underwent surgical re-treatment for BPH (TURP). All patients in both groups who underwent the procedure under local anesthesia were discharged on postoperative day 1.

## 4. Discussion

One of the benefits of the PUL is that it does not require spinal or general anesthesia and can be performed under local anesthesia. This is especially relevant in older patients, in whom perioperative care may be associated with an increased risk due to multiple comorbidities and decreased physical health [18]. In this study, we compared the clinical efficacy of the PUL in healthy individuals and those with multiple comorbidities. The results of this study showed that the PUL had a high treatment failure rate, and that improvement of all functional outcomes was significantly lower in patients with high comorbidities than in healthy individuals. In addition, in these patient groups, IPSS and QoL improvements after surgery were not maintained for more than 1 year, and in the case of Qmax, for no more than 6 months. To our knowledge, this is the first study to report the clinical outcomes of PUL in patients at high risk for general anesthesia due to multiple comorbidities.

Several authors have studied age-related clinical outcomes after BPH surgery. Hakenberg et al. (2003) analyzed preoperative factors for predicting symptom improvement after TURP. They demonstrated that age was the only independent predictor of flow rate and symptom improvement [19]. Similar results were reported by Giovanni et al. (2013); they found that treatment success after TURP, in terms of freedom from catheter dependence, was affected by age [20]. In particular, treatment failure rate was 24% in men aged ≥80 years, which increased to 57% in those aged ≥85 years. Another group reported an age-stratified assessment of outcomes after HoLEP and found that patients aged ≥80 years had less improvement in Qmax compared to younger age groups during the 1-year follow-up [21]. These results suggest that older patients who are considered candidates for surgical treatment in a clinical context should be further evaluated before a surgical treatment is recommended.

Additionally, several authors have reported the effects of comorbidities on the clinical outcomes of TURP. Sener et al. (2015) demonstrated that patients with a metabolic syndrome showed less symptom improvement after TURP [22]. Recently, Zang et al. (2019) reported that men with a lower number of comorbidities had improved outcomes after TURP, regardless of age [23]. Another study found that patients with a higher ASA score tended to have greater benign prostatic obstruction-related complications after TURP than patients with lower ASA [24]. These results suggesting an association between preoperative comorbidity and worse clinical outcomes after TURP are similar to those observed in the PUL procedure. Indeed, because the current study compared healthy individuals to patients with an ASA score of ≥3 with multiple and/or severe comorbidities, these differences in clinical outcomes were clearly observed.

Unlike clinical efficacy, no relationship between severity of comorbidities and adverse events after PUL was noted in this study. This is a procedural advantage of the PUL, similar to other minimally invasive treatments. In addition, previous studies have reported that complications associated with PUL are significantly lower than those associated with TURP [14]. All patients in our study underwent a PUL under intravenous or local anesthesia and were discharged on postoperative day 1. The adverse events were mild, transient, and resolved within the first postoperative week. The most common adverse events in patients were dysuria and hematuria, which are similar to those reported in previous studies on the PUL [25,26,27,28].

Another benefit of the PUL is that it has less impact on ejaculatory function, as it preserves the bladder neck. Many transurethral prostate resection procedures are associated with reduced ejaculatory function and sometimes cause de novo erectile dysfunction. Studies on TURP and holmium laser surgery have reported a high incidence of post-procedural ejaculatory dysfunction, reaching up to 80% [29]. De novo postoperative ED rates after such surgeries have been reported to range from 3 to 14% [30]. Due to these complications associated with traditional procedures, the PUL has been gaining traction as a minimally invasive, ejaculation-preserving alternative to TURP. McVary et al. (2015) and Perera et al. (2014) reported that PUL improves LUTSs and preserves sexual function, particularly ejaculatory function [26,31]. Our study showed similar results, since sexual parameters, including erectile and ejaculatory function as measured by IIEF-5 and MSHQ-EjD questionnaires, remained stable post-procedure, even in patients with comorbidities.

Regarding clinical outcomes at the 24-month follow-up in healthy individuals, the current study showed similar results compared with recently introduced minimally invasive surgical treatments, including Aquablation, steam injection (Rezum), and prostate artery embolization (PAE). Wang et al. (2015) investigated the clinical efficacy of PAE in patients with a prostate volume >80 g [32]. They reported significant improvements in IPSS (a decrease of 17 points) and Qmax (an increase of 6mL/s) at 24 months after PAE. Roehrborn et al. (2017) conducted a multicenter randomized controlled trial to evaluate the efficacy of Rezum at 2 years post-therapy [33]. They reported that IPSS improved by 11.2 points and Qmax increased by 4.2 mL/s from the baseline. Gilling et al. (2019) reported the 2-year outcome of Aquablation compared TURP [34]. They reported that IPSS improved by 14.7 points and Qmax increased by 11.2 mL/s at the 2-year mark. Although several studies demonstrated the clinical efficacy of minimally invasive surgical treatments in patients with BPH, no study has yet reported the clinical outcomes of patients with multiple comorbidities, as in the current study. Therefore, clinical outcomes for the PUL in patients with multiple comorbidities, as described in this study, were not comparable to other minimally invasive treatments.

The main limitation of this study is that it was performed at a single center and used a retrospective cohort design. Our follow-up protocol was extensive and, while structured and homogenous in design, some patients were lost during the follow-up period. Additionally, the number of participants in this study was relatively small, which may not have provided adequate statistical power. Nevertheless, significant differences between the two groups were observed in postoperative clinical outcomes, such as IPSS, QoL, and Qmax. Another limitation is that this study was initiated with the introduction of the PUL, and the learning curve for the procedure may have affected the results. More rigorous prospective controlled trials, with a large number of subjects, are needed in the future.

Despite these limitations, this study suggests that comorbidity status should be considered when evaluating the expected benefits of a PUL in preoperative counseling. Although PUL is a conceivable alternative in patients at a high risk for general anesthesia due to comorbidities, it should be recognized that severe and multiple comorbidities can lead to treatment failure after PUL. Future studies based on new concepts introduced in recent years should aim to determine whether other minimally invasive treatments are indeed beneficial for these patients.

## Figures and Tables

**Figure 1 jcm-11-01928-f001:**
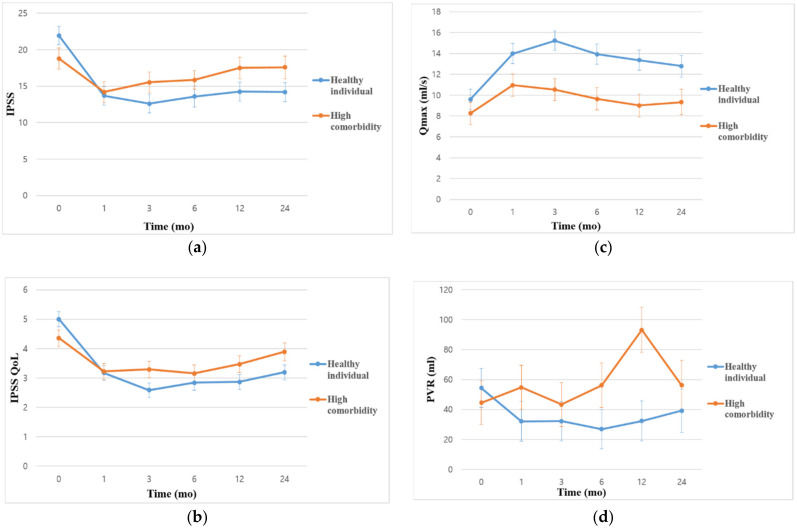
A generalized linear mixed model shows the trend of outcomes following treatment with prostatic urethral lift for healthy individuals and those with a high number of comorbidities in terms of (**a**) International Prostate Symptom Score (IPSS), (**b**) IPSS quality of life (QoL), (**c**) maximum urinary flow rate (Qmax), and (**d**) post void residual volume (PVR) over the course of 24 months. Values are plotted as least squares mean with standard error.

**Table 1 jcm-11-01928-t001:** Baseline and perioperative clinical characteristics in both groups.

Characteristic	Healthy Individual (n = 36)	High Comorbidity (n = 30)	*p*-Value
Age (years), mean ± SD	60.7 ± 5.3	75.3 ± 8.1	<0.001
Total prostate volume (cm^3^), mean ± SD	50.6 ± 18.8	46.5 ± 15.4	0.102
PSA (ng/mL), mean ± SD	2.0 ± 1.3	2.5 ± 2.3	0.257
IPSS-Total, mean ± SD	21.4 ± 5.2	19.4 ± 5.9	0.143
QoL, mean ± SD	4.9 ± 1.1	4.5 ± 1.0	0.209
Qmax (mL/s), mean ± SD	9.8 ± 4.1	8.0 ± 4.6	0.105
Post void residual volume (mL), mean ± SD	54.4 ± 82.4	44.7 ± 38.2	0.552
IIEF-5, mean ± SD	15.8 ± 6.6	9.4 ± 6.8	0.001
MHSQ-function, mean ± SD	11.1 ± 3.8	9.8 ± 3.3	0.246
MHSQ-bother, mean ± SD	1.6 ± 1.3	1.4 ± 0.7	0.764
Comorbidities			
Diabetes, number (%)	4 (11.1)	20 (66.7)	<0.001
Cardiovascular disease, number (%)	0 (0)	14 (46.7)	<0.001
Chronic obstructive pulmonary disease, number (%)	0 (0)	16 (56.7)	<0.001
Catheter in place at procedure, number (%)	2 (5.6)	4 (13.3)	0.399
Types of anesthesia			<0.001
Local, number (%)	6 (16.7)	30 (100)	
General, number (%)	30 (83.3)	0 (0)	
PUL implants, mean ± SD	2.8 ± 0.8	2.7 ± 0.7	

SD, standard deviation; PSA, prostate-specific antigen; IPSS, International Prostate Symptom Score; QoL, Quality of Life; Qmax, maximum urinary flow rate; IIEF-5, International index of erectile function; MSHQ-EjD, Male sexual health questionnaire for ejaculatory dysfunction, short-form version (MSHQ-EjD); PUL, prostatic urethral lift.

**Table 2 jcm-11-01928-t002:** Paired outcomes after prostatic urethral lift in both groups.

Variable	1 Month		3 Months		6 Months		12 Months		24 Months	
	Healthy Individuals	High Comorbidity	Healthy Individuals	High Comorbidity	Healthy Individuals	High Comorbidity	Healthy Individuals	High Comorbidity	Healthy Individuals	High Comorbidity
IPSS, n	31	30	34	30	32	30	30	26	23	18
Baseline, mean ± SD	22 ± 5.0	19.4 ± 5.9	21.5 ± 5.1	19.4 ± 5.9	21.7 ± 5.2	19.4 ± 5.9	22.0 ± 5.1	19.7 ± 6.3	22.5 ± 5.3	17.0 ± 4.5
Follow-up, mean ± SD	13.4 ± 8.3	14.8 ± 6.9	12.3 ± 6.2	16.1 ± 8.7	13.8 ± 5.8	16.5 ± 7.6	14.6 ± 6.2	17.2 ± 7.1	13.9 ± 6.6	14.8 ± 4.7
Change, mean ± SD	−8.6 ± 7.1	−4.7 ± 6.7	−9.2 ± 6.3	−3.3 ± 8.5	−7.9 ± 6.6	−2.9 ± 7.1	−7.4 ± 6.6	−2.6 ± 3.1	−8.6 ± 6.1	−2.2 ± 4.5
Change *p*-value	<0.001 *	0.001 *	<0.001 *	0.045 *	<0.001 *	0.03 *	<0.001 *	0.001 *	<0.001 *	0.054 *
Comparison *p*-value	0.032		0.002		0.006		0.001		0.001	
QOL, n	31	30	34	30	32	30	30	26	23	18
Baseline, mean ± SD	5.0 ± 1.1	4.5 ± 1.0	4.9 ± 1.1	4.5 ± 1.0	4.9 ± 1.1	4.5 ± 1.0	4.9 ± 1.2	4.5 ± 1.0	4.8 ± 1.2	4.1 ± 0.9
Follow-up, mean ± SD	3.1 ± 1.7	3.4 ± 1.4	2.5 ± 1.4	3.5 ± 1.7	2.8 ± 1.4	3.3 ± 1.3	2.9 ± 1.4	3.5 ± 1.3	3.0 ± 1.3	3.6 ± 0.9
Change, mean ± SD	−1.9 ± 1.6	−1.1 ± 1.4	−2.4 ± 1.4	−1.1 ± 1.8	−2.1 ± 1.6	−1.2 ± 1.5	−2.0 ± 1.4	−1.2 ± 1.4	−1.8 ± 1.3	−0.6 ± 1.1
Change *p*-value	<0.001 ^†^	0.001 ^†^	<0.001 ^†^	0.004 ^†^	<0.001 ^†^	0.001 ^†^	<0.001 ^†^	<0.001 ^†^	<0.001 ^†^	0.045 ^†^
Comparison *p*-value	0.039		0.002		0.022		0.038		0.002	
Qmax (mL/s), n	28	30	31	30	30	30	26	26	18	18
Baseline, mean ± SD	9.7 ± 4.2	8.0 ± 4.6	10.1 ± 4.3	8.0 ± 4.6	10.3 ± 4.2	8.0 ± 4.6	9.1 ± 3.9	8.4 ± 4.8	10.0 ± 3.9	9.5 ± 5.3
Follow-up, mean ± SD	14.0 ± 7.5	10.7 ± 6.0	15.9 ± 6.2	10.3 ± 5.2	14.4 ± 6.9	9.4 ± 5.7	12.8 ± 5.8	9.2 ± 4.7	13.3 ± 6.2	10.4 ± 4.7
Change, mean ± SD	4.3 ± 5.7	2.7 ± 4.6	5.8 ± 5.0	2.3 ± 3.8	4.1 ± 5.2	1.4 ± 4.3	3.8 ± 4.1	0.8 ± 3.1	3.3 ± 4.8	0.9 ± 4.0
Change *p*-value	<0.001 *	0.004 ^†^	<0.001 *	0.002 ^†^	<0.001 *	0.137 ^†^	<0.001 *	0.469 ^†^	0.009 *	0.325 ^†^
Comparison *p*-value	0.242		0.003		0.028		0.006		0.112	
PVR, n	28	30	31	30	30	28	25	26	17	20
Baseline, mean ± SD	53.6 ± 77.3	44.7 ± 38.2	46.5 ± 73.7	44.7 ± 38.2	49.3 ± 75.2	36.4 ± 22.6	58.0 ± 80.5	49.2 ± 39.1	65.9 ± 95.5	36.0 ± 20.6
Follow-up, mean ± SD	34.8 ± 32.1	54.9 ± 59.7	30.0 ± 53.4	43.5 ± 38.1	23.0 ± 24.3	37.1 ± 29.7	36.4 ± 42.2	92.3 ± 160.4	44.7 ± 92.7	32.5 ± 29.8
Change, mean ± SD	−18.8 ± 79.1	10.3 ± 53.1	−16.5 ± 59.3	−1.2 ± 36.6	−26.3 ± 67.1	0.7 ± 30.7	−21.6 ± 67.4	43.1 ± 128.5	−21.2 ± 78.7	−3.5 ± 36.4
Change *p*-value	0.475 ^†^	0.841 ^†^	0.217 ^†^	0.509 ^†^	0.061 ^†^	0.498 ^†^	0.286 ^†^	0.651 ^†^	0.483 ^†^	0.153 ^†^
Comparison *p*-value	0.104		0.233		0.053		0.03		0.403	
IIEF-5, n	19	6	23	20	22	16	23	14	21	14
Baseline, mean ± SD	16.9 ± 5.8	16.7 ± 1.0	16.4 ± 6.0	9.4 ± 6.8	16.3 ± 6.2	11.4 ± 6.2	16.1 ± 6.1	13.0 ± 4.6	16.5 ± 6.1	13.0 ± 4.6
Follow-up, mean ± SD	15.6 ± 6.9	16.7 ± 1.0	15.5 ± 6.1	9.8 ± 6.6	16.8 ± 5.6	11.9 ± 5.5	16.4 ± 6.2	13.3 ± 3.5	16.2 ± 5.7	13.0 ± 5.2
Change, mean ± SD			−1.3 ± 3.6	0.4 ± 2.2	0.5 ± 4.0	0.5 ± 2.6	0.3 ± 4.4	0.3 ± 2.6	−0.2 ± 3.8	0.0 ± 1.8
Change *p*-value	0.033 ^†^	1.00 ^†^	0.199 ^†^	0.497 ^†^	0.681 ^†^	0.691 ^†^	0.839 ^†^	0.874 ^†^	0.380 ^†^	0.794 ^†^
Comparison *p*-value			0.065		0.985		0.83			
MHSQ-function, n	15	6	15	16	10	12	19	12		
Baseline, mean ± SD	10.5 ± 4.0	9.0 ± 1.5	12.1 ± 3.1	9.7 ± 3.3	12.6 ± 3.0	9.3 ± 3.5	11.2 ± 3.8	9.3 ± 3.5		
Follow-up, mean ± SD	9.3 ± 5.1	9.0 ± 1.5	11.1 ± 3.6	9.2 ± 3.3	10.7 ± 2.9	9.2 ± 3.3	10.9 ± 2.6	9.2 ± 3.0		
Change, mean ± SD			−1.1 ± 3.3	−0.6 ± 1.1	−2.8 ± 4.1	0.1 ± 3.2	−1.0 ± 4.4	−0.2 ± 2.0		
Change *p*-value	0.447 ^†^	0.914 ^†^	0.234 ^†^	0.058 ^†^	0.052 ^†^	0.959 ^†^	0.528 ^†^	0.776 ^†^		
Comparison *p*-value			0.167		0.081		0.544			
MHSQ-bother, n	15	6	15	16	10	12	19	12		
Baseline, mean ± SD	1.6 ± 1.5	1.7 ± 0.5	1.1 ± 0.9	1.5 ± 0.7	0.9 ± 0.9	1.5 ± 0.8	1.4 ± 1.3	1.4 ± 0.8		
Follow-up, mean ± SD	1.7 ± 1.8	1.7 ± 0.5	1.2 ± 1.3	1.5 ± 0.7	1.4 ± 1.3	1.8 ± 0.9	1.6 ± 1.3	1.4 ± 0.9		
Change, mean ± SD			0.1 ± 0.9	0.0 ± 0.5	0.0 ± 2.1	0.3 ± 0.5	0.1 ± 1.1	0.2 ± 0.4		
Change p-value	0.829 ^†^	1.00 ^†^	0.589 ^†^	1.00 ^†^	0.301 ^†^	0.046 ^†^	0.357 ^†^	1.00 ^†^		
Comparison *p*-value			0.618		0.625		0.838			

SD, standard deviation; PSA, prostate-specific antigen; IPSS, International Prostate Symptom Score; QoL, Quality of Life; Qmax, maximum urinary flow rate; PVR, post-void residual volume; IIEF-5, International index of erectile function, MSHQ-EjD; Male sexual health questionnaire for ejaculatory dysfunction, short-form version (MSHQ-EjD). * Paired Student’s *t*-test, ^†^ Wilcoxon signed-rank test.

**Table 3 jcm-11-01928-t003:** Treatment-related adverse events and re-intervention following treatment with PUL.

Adverse Event	Healthy Individuals (n = 36)	High Comorbidity (n = 30)	*p*-Value *
Dysuria, n (%)	6 (16.7)	4 (13.3)	0.745
Hematuria, n (%)	4 (11.1)	5 (16.7)	0.721
Prostate abscess, n (%)	1 (2.8)	0	
Treatment failure, n (%)	3 (8.3)	11 (36.7)	0.007

PUL = prostatic urethral lift. * Fisher’s exact test.

## Data Availability

Not applicable.

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
