# Peer review of "Is Prostate Urethral Lift Effective in Patients with Multiple Comorbidities?"

_jcm, 2022, doi:10.3390/jcm11071928_

Round 1

Reviewer 1 Report

The manuscript is a valuable research. Surgical management of prostatic enlargement and lower urinary tract symptoms (LUTS) is one of the main activities in urology. The article gives a good overview of the issue. However, some points may be reviewed.

The introduction is a description of the state of art. However, there is some information that requires revision. Authors state that  “The initial therapeutic option for LUTS is watchful waiting, followed by pharmacotherapy with alpha-blockers or 5-alpha reductase (5-ARI).” Management of LUTs must be based on symptoms severity risk of progression, and watchful waiting is not the initial therapeutic option in many cases. Moreover, currently, more drugs are available, such as those for filling symptoms. Better organization according to EAU and AUA guidelines is recommended.

Surgical treatment of HBP is not associated with “hospitalized for a relatively long period and may lead to complications, such as hematuria, urinaryincontinence, and urethral stricture” Please add data about the mean hospitalization periods and the percentage of complications with other techniques.

Authors select patients who desire “preserve ejaculatory function”. Please add information about another method that preserves ejaculation, such as Aquabeam.

Please describe the intravenous sedation when it was used.

A discussion comparing the results with other published studies is recommended, including emerging technologies such as Rezum, Aquabeam, prostatic artery embolization.

Author Response

Response to Reviewer 1 Comments  

Point 1: The introduction is a description of the state of art. However, there is some information that requires revision. Authors state that  “The initial therapeutic option for LUTS is watchful waiting, followed by pharmacotherapy with alpha-blockers or 5-alpha reductase (5-ARI).” Management of LUTs must be based on symptoms severity risk of progression, and watchful waiting is not the initial therapeutic option in many cases. Moreover, currently, more drugs are available, such as those for filling symptoms. Better organization according to EAU and AUA guidelines is recommended.

Response 1: Thank you for your comment. We revised the sentence as you recommend.

Point 2: Surgical treatment of HBP is not associated with “hospitalized for a relatively long period and may lead to complications, such as hematuria, urinaryincontinence, and urethral stricture” Please add data about the mean hospitalization periods and the percentage of complications with other techniques.

 Response 2: Thanks for good point. As you pointed out, we deleted and revised vague expressions and content that may be misunderstood. In accordance with the original purpose, the representative complications of TURP and their rates added to the manuscript.

Point 3: Authors select patients who desire “preserve ejaculatory function”. Please add information about another method that preserves ejaculation, such as Aquabeam.

Response 3: Thanks for the recommendation. However, other minimally invasive treatments such as Aquablation and Rezum have not yet been approved in Korea, so the procedure could not be performed. Information related to this has been added in the manuscript.

Point 4:  Please describe the intravenous sedation when it was used.

Response 4: As you recommend, we added the information.

Point 5: A discussion comparing the results with other published studies is recommended, including emerging technologies such as Rezum, Aquabeam, prostatic artery embolization.

Response 5: Thank you for your great opinion. As recommended, we have added comparisons with recent emerging minimal invasive treatments. Unfortunately, there were no studies targeting patients with multiple comorbidities as current study, so comparison clinical outcomes for patients with multiple comorbidities was not possible.

Reviewer 2 Report

The concept of the study is great. It aims to see the efficacy of PUL in patients who have comorbidities such as diabetes etc. However, the title is completely misleading and authors need to modify the title- reading the title only makes it feel that the authors are trying to compare PUL with local anesthesia vs general but that is not the aim of this study. The title needs to be changed completely. 

Author Response

Response to Reviewer 2 Comments

Point: The concept of the study is great. It aims to see the efficacy of PUL in patients who have comorbidities such as diabetes etc. However, the title is completely misleading and authors need to modify the title- reading the title only makes it feel that the authors are trying to compare PUL with local anesthesia vs general but that is not the aim of this study. The title needs to be changed completely. 

Response: We totally agree with you. We revised the title of the study as you recommend.

Round 2

Reviewer 2 Report

Authors have revised the title and manuscript as per the prior review suggestions.